# Self-assembly of multi-stranded perylene dye J-aggregates in columnar liquid-crystalline phases

Stefanie Herbst[1], Bartolome Soberats[2,3], Pawaret Leowanawat[1], Matthias Stolte[1,2], Matthias Lehmann[1,2,3] & Frank Würthner [1,2,3]

Many discoid dyes self-assemble into columnar liquid-crystalline (LC) phases with packing arrangements that are undesired for photonic applications due to H-type exciton coupling. Here, we report a series of crystalline and LC perylene bisimides (PBIs) self-assembling into single or multi-stranded (two, three, and four strands) aggregates with predominant J-type exciton coupling. These differences in the supramolecular packing and optical properties are achieved by molecular design variations of tetra-bay phenoxy-dendronized PBIs with two N–H groups at the imide positions. The self-assembly is driven by hydrogen bonding, slipped π–π stacking, nanosegregation, and steric requirements of the peripheral building blocks. We could determine the impact of the packing motifs on the spectroscopic properties and demonstrate different J- and H-type coupling contributions between the chromophores. Our findings on structure–property relationships and strong J-couplings in bulk LC materials open a new avenue in the molecular engineering of PBI J-aggregates with prospective applications in photonics.

[1] Institut für Organische Chemie, Universität Würzburg, Am Hubland, 97074 Würzburg, Germany. [2] Center for Nanosystems Chemistry (CNC), Universität Würzburg, Theodor-Boveri-Weg, 97074 Würzburg, Germany. [3] Bavarian Polymer Institute (BPI), Universität Würzburg, Theodor-Boveri-Weg, 97074 Würzburg, Germany. Correspondence and requests for materials should be addressed to M.L. (email: matthias.lehmann@uni-wuerzburg.de) or to F.Wür. (email: wuerthner@uni-wuerzburg.de)

Perylene bisimide (PBI) dyes are attracting a great deal of attention not only for their outstanding photophysical and optoelectronic properties[1–3], but also for their self-assembly properties[4–7]. They exhibit a strong tendency to aggregate via π–π interactions and are therefore widely used to fabricate supramolecular structures in bulk and in solution[4–10]. The control of the self-assembly processes of PBI dyes into structures with tailored functions has become a paradigm in the field of supramolecular materials science. To date, PBI aggregates have been applied as semiconductors for solar cells[11,12] and field effect transistors[2,13], and as functional mimics of biological photosystems[14,15]. In general, molecular and supramolecular engineering has been exploited to adjust the exciton coupling in PBI assemblies, which commonly lead to H- or J-type aggregates[16–21]. The latter type of assembly was first reported by Jelley[22] and Scheibe[23] and is of special interest for photonics due to intriguing photophysical properties, i.e., intense bathochromically shifted absorption bands with respect to their monomers, high fluorescence quantum yields, and relatively long exciton lifetimes[24]. Previously, we have reported hierarchically organized PBI J-aggregates that form fibers in solution and show intense fluorescence and exciton migration over distances up to 100 nm[25,26]. These are attractive for energy transport applications; however, implementation in devices demands the further organization of these fibrous aggregates in the bulk material.

Toward this goal, liquid crystals have proven to be a suitable platform to develop functional liquid-crystalline (LC) materials[27–31], and for the self-organization of PBI dyes[4,32–38]. In particular, Percec and co-workers have recently exploited the functionalization of PBIs at the imide position with different dendrons to obtain LC columnar assemblies[34–37]. Typically, imide dendronization of PBIs enables the formation of columnar assemblies formed via cofacial π–π interactions between chromophores. Many works have focused on the variation of PBI molecular design, for example changing the nature of the dendrons (first, second, and third generation, perfluorinated, etc.) and the alkyl spacers between dendrons and cores[34–36]. The importance of molecular design is clearly exemplified in recent examples, where uncommon organizations in PBI liquid crystals were found[37,38]. In this vein, we recently reported an unprecedented organization of PBIs into LC triple-stranded helical columnar structures in which the PBI cores are oriented parallel to the columnar axis[38]. This material was subsequently utilized in photonic microcavities[39]. Our molecular design consists of a tetra-bay functionalized MEH-PBI with 1,2-ethylhexyl (EH) substituents and free NH groups at the imide positions. These molecules self-organize via hydrogen bonds (H-bonds) and slipped π–π interactions into strongly coupled J-aggregates. In the next step of our research, we aim to understand the driving forces involved in this assembly and to control the number of strands in the columns via molecular engineering of the PBI building blocks. We envisioned that such control of the assembly would permit the fine-tuning of the excitonic interactions between dyes and the photophysical properties of the J-aggregates.

Herein, we report on the self-assembly behavior of tetra-bay dendronized PBIs 1–4 (Fig. 1a), which self-organize via H-bonding and/or slipped π–π interactions (Fig. 1b) into one-, two-, three-, or four-stranded J-aggregates within crystalline and columnar LC phases (Fig. 1c). Molecular design and the attachment position of the dendrons at the phenoxy-PBI scaffolds dictates organization into different numbers of PBI strands (Fig. 1). The twist in the PBI cores and the steric demand of the dendrons play a key role in the columnar assembly modes, which in turn have an impact on the exciton couplings between dyes and on spectroscopic properties such as absorption and emission maxima of the J-aggregates. These results are of great importance

for the fine-tuning of properties of tailor-made materials for photonic applications as it was recently demonstrated for MEH-PBI in imprinted LC pillar microcavities[39].

## Results

**Material design and LC behavior of PBIs 1–4.** The basic structure of the PBI compounds consists of a 1,6,7,12-tetraphenoxy PBI with four dendrons attached to the phenoxy spacers and two unblocked NHs at the imide positions (Fig. 1a). The present series of compounds (PBIs 1–4) was designed by using four different phenoxy spacers (ortho-, meta-, para-hydroxyl-, and 2-hydroxy-6-methyl phenoxy) and 3,4,5-tridodecyloxybenzoate dendrons (Fig. 1a and Supplementary Fig. 1). PBI 1 has the wedge-shaped groups at the ortho- position of the phenoxy spacers, while PBIs 2 and 3 bear them at the meta- and para-positions, respectively (Fig. 1a). In PBI 4, the dendronized benzoates are also located at the meta- positions, but they possess an additional methyl group at the ortho- position of the phenoxy spacer to reduce the conformational flexibility of the phenoxy substituents (Fig. 1a)[40]. Bay tetraphenoxy-substituted PBIs exhibit a ~25˚ twist of the PBI core due to steric demands of the phenoxy units at the bay positions[25].

PBIs 1–4 were prepared according to our previously described methodology (Supplementary Fig. 1)[25,38]. While PBI 1 was obtained as a crystalline red powder, the PBIs 2–4 where obtained as waxy dark blue solids. The phase characterization of the tetra-bay-substituted PBIs 1–4 was performed by polarizing optical microscopy (POM), differential scanning calorimetry (DSC), and wide and middle angle X-ray scattering (WAXS and MAXS) analysis (Fig. 2 and Supplementary Figs. 2–10). Table 1 summarizes the phase behavior for the four PBI derivatives.

Figure 2 illustrates the X-ray patterns and POM images of PBIs 1–4 at different temperatures. PBI 1 exhibits a crystalline phase evidenced by the formation of needle-like crystals and the non-fluid nature of the sample (Fig. 2a inset). The crystalline nature of the sample is further confirmed by the high number of reflections in the X-ray pattern (Fig. 2a). The most prominent reflections in the X-ray pattern of PBI 1 can be indexed with Miller indices 010, 110, 1−10, 200, 001, 10−1, 101, 210, 020, 2−1−1, 220, 021, 002, and 102 (Fig. 2a) corresponding to a triclinic lattice with parameters $a = 41.3$ Å, $b = 28.2$ Å, and $c = 20.4$ Å ($\alpha = 96°$, $\beta = 97°$, $\gamma = 90°$). In contrast, PBIs 2–4 exhibit columnar LC phases as evidenced by the fluidity of the samples as well as POM observations and X-ray measurements (Fig. 2b–d). PBI 2 exhibits both a columnar rectangular (Col$_r$) phase from 20 to 210 °C and a columnar hexagonal (Col$_h$) phase up to 255 °C (Table 1). The transition between the Col$_r$ phase and the Col$_h$ phase can be clearly observed by DSC experiments with a transition enthalpy of +5.5 kJ/mol and X-ray scattering experiments (Supplementary Figs. 2–6). In the case of the low temperature columnar phase, eight reflections are observed in the X-ray pattern of PBI 2 (Fig. 2b). These signals correspond to the Miller indices 110, 200, 020, 220, 400, 130, and 510 of a Col$_r$ phase ($a = 69.7$ Å, $b = 47.8$ Å). In contrast, by increasing the temperature to 250 °C, the typical pattern of a Col$_h$ phase ($a = 39.9$ Å) is observed by X-ray experiments (Supplementary Figs. 5 and 6). Para-functionalized PBI 3 exhibits an LC phase between 20 and 240 °C (Supplementary Fig. 2). The reflections on the equator (100, 110, 200, 210, and 300) were indexed according to a Col$_h$ lattice with unit cell parameter $a = 40.9$ Å (Fig. 2c). Compound 4 also reveals a columnar LC phase from −40 °C up to 319 °C (Table 1) exhibiting a pseudo-focal-conic texture under crossed polarizers (Fig. 2d inset). Five reflections on X-ray pattern were indexed with Miller indices 100, 110, 200, 210, and 300 (Fig. 2d) according to a Col$_h$ lattice with $a = 32.3$ Å (Table 1).

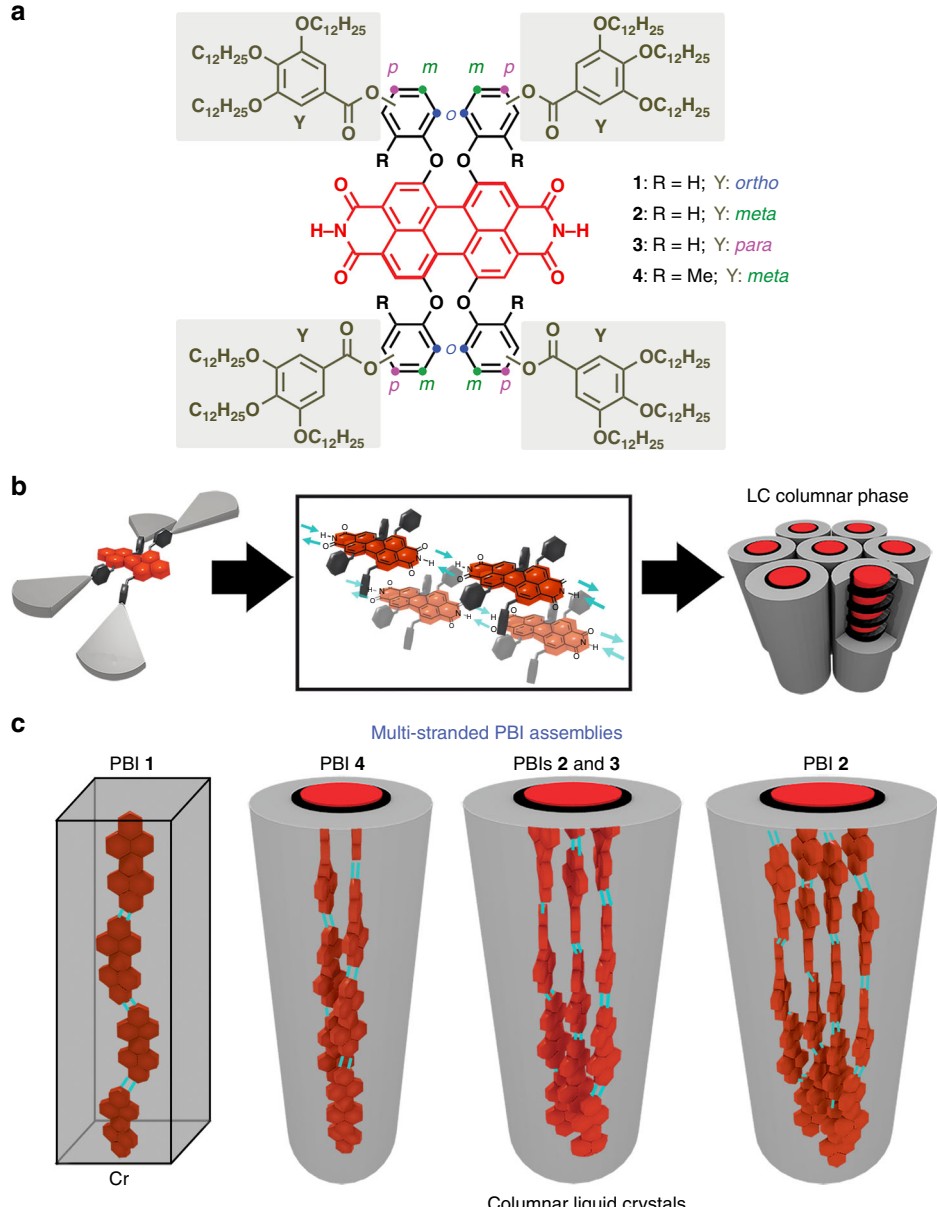

**Fig. 1** Molecular design and self-assembled structures of the PBIs **1–4**. **a** Chemical structures of PBIs **1–4**. **b** Schematic illustration of the self-assembly of the PBIs via self-complementary hydrogen bonding (H-bonding) and π–π interactions. **c** Schematic illustration of the different columnar assemblies composed of one, two, three, and four strands. The helicity in the columnar assemblies of PBIs **2–4** has been chosen to be (P) for graphical representation. Please note that PBIs **1–4** are racemic and therefore (P) and (M) helices may coexist in the corresponding columnar phases

From the above-mentioned experiments, it is clear that all PBI derivatives self-assemble in ordered, anisotropic structures independent of the substitution pattern at the phenoxy spacers (Fig. 1 and Table 1). PBI **1** forms a crystalline phase while PBIs **2–4** generate LC phases. However, striking differences in these assemblies were deduced from UV–vis spectra (in solid state) and the estimated number of molecules that compose the columnar strata in the different PBI LC assemblies (Table 1 and Supplementary Methods). The UV–vis thin film spectra in the LC phase of solution-cast PBIs **2–4** show bathochromically shifted bands with respect to the PBI monomeric bands in $CH_2Cl_2$ (Supplementary Fig. 11). This is consistent with the formation of strongly coupled J-aggregates[21–26]. However, the absorption spectrum of PBI **1** in the crystalline state exhibits only a slight bathochromic shift and a retained vibronic fine structure of the PBI absorption band (Supplementary Fig. 12a) that

suggests a weak coupling between the dyes' transition dipole moments. The optical features of PBIs **1–4** are analyzed in detail later in this paper.

The number of molecules per columnar slice for all the LC PBIs was estimated from X-ray data using extrapolated densities[41,42] and the experimental columnar height of 13.8–14.2 Å that corresponds approximately to the length of a single PBI dye (Supplementary Tables 1 and 4). Interestingly, we found that the columnar slices of the $Col_h$ phases of PBIs **3** and **4** contain three and two molecules, respectively. The lower temperature $Col_r$ phase of PBI **2** has four molecules per columnar slice while the higher temperature $Col_h$ phase consists of three molecules per slice (Table 1). PBI **1** shows a crystalline phase, which best fits a triclinic unit cell filled with four molecules. These results suggest that the four PBI isomers present different arrangements despite similar chemical structures. These

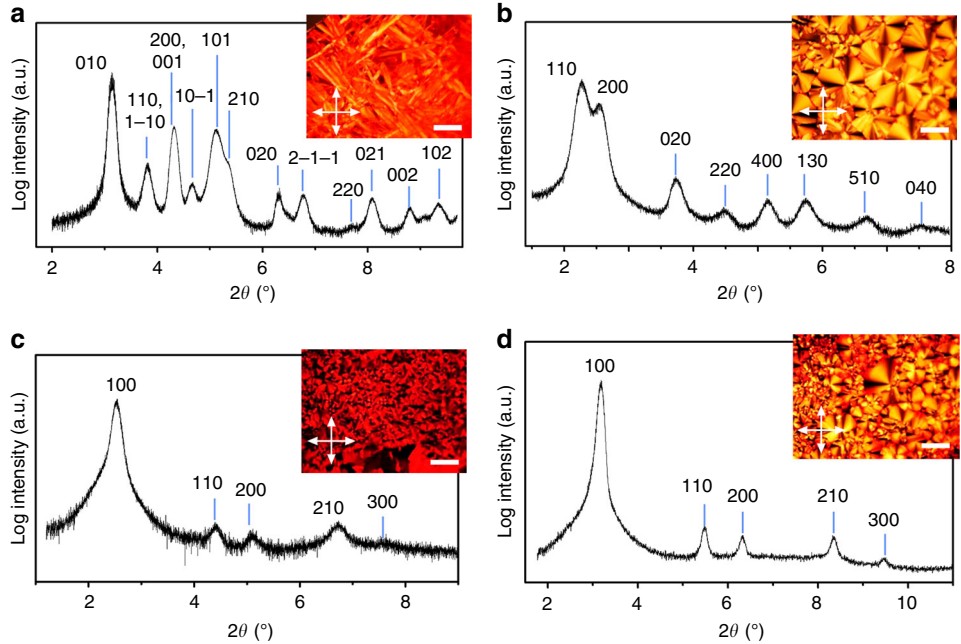

**Fig. 2** X-ray patterns and POM images of PBIs **1–4**. Integrated intensities of the MAXS powder pattern of PBI **1** at 136 °C. Inset shows the POM image of PBI **1** at 152 °C (**a**). Integrated intensities along the equator of the MAXS patterns of aligned fibers of PBI **2** at 160 °C (**b**), PBI **3** at 180 °C (**c**), and PBI **4** at 224 ° C (**d**). Insets show the POM images of PBI **2** at 240 °C (**b**), PBI **3** at 210 °C (**c**), and PBI **4** at 270 °C (**d**) with the respective scale bars indicating a length of 100 μm. Arrows indicate the direction of analyzer and polarizer. The corresponding MAXS patterns are shown in Supplementary Figs. 3, 7, 9

| Table 1 LC behavior and lattice parameter of PBIs 1–4 | | | |
|---|---|---|---|
| **PBI** | **Transition temp. [°C]** | **Lattice parameter [Å]** | **Number of molecules per columnar stratum[a]** |
| **1** | H: 20[b] Cr 150 Iso<br>C: Iso 140 Cr 20[b] | (Cr) $a = 41.3$, $b = 28.2$, $c = 20.4$, $\alpha = 96°$, $\beta = 97°$, $\gamma = 90°$ | –[c] |
| **2** | H: 20[b] Col$_r$ 210 Col$_h$ 252 Iso<br>C: Iso 253 Col$_h$ 186 Col$_r$ 20[b] | (Col$_r$) $a = 69.7$, $b = 47.8$<br>(Col$_h$) $a = 39.9$ | 4<br>3 |
| **3** | H: 20[b] Col$_h$ 235 Iso<br>C: Iso 237 Col$_h$ 20[b] | (Col$_h$) $a = 40.2$ | 3 |
| **4** | H: Cr −47[d] Cr −25 Col$_h$ 279 Iso<br>C: Iso 282 Col$_h$ −20 Cr −44 Cr | (Col$_h$) $a = 32.3$ | 2 |

H 2nd heating cycle, C 1st cooling cycle, Cr crystalline, Iso isotropic liquid, Col$_h$ columnar hexagonal, Col$_r$ columnar rectangular
[a]The number of molecules per stratum was calculated for the columnar heights determined from X-ray experiments (13.8–14.2 Å). Further details about these calculations are in Supplementary Methods
[b]Starting temperature of DSC measurement
[c]The triclinic unit cell of PBI **1** contains four molecules. The number of molecules per repeating unit of the strand in the crystalline phase could not be calculated from the X-ray data, but FT-IR and UV–vis spectroscopy suggest that the assembly consists of single strands (vide infra)
[d]Peak temperature

surprising results prompted us to study these columnar assemblies in more detail and to determine the implication of structural differences on the optical properties of the materials.

**Structural analysis and modeling of the PBI assemblies**. In order to obtain a more detailed picture of the self-assembled structures of PBIs **1–4**, we first focused on the elucidation of the relative orientation of the chromophores in the assemblies. For the crystalline structure of PBI **1**, solid state UV–vis spectra (Supplementary Fig. 12) suggest that the PBI cores do not strongly couple to each other. Temperature-dependent FT-IR experiments reveal a shift in the NH vibration band when the sample melts, which is associated with the cleavage of intermolecular H-bonds (Supplementary Fig. 13). In the case of PBI **1**, the exact arrangement of the dyes could not be determined by the X-ray studies, but UV–vis and FT-IR spectroscopy experiments are only consistent with a hydrogen-bond directed assembly based on a single strand. This can be rationalized by the steric effects of the dendrons at the *ortho-* positions, which prevent π–π

interactions in between PBIs and lead to an H-bonded single strand organization via the self-complementary NH–CO functionality (Fig. 1c).

The anisotropic features of LC compounds (**2–4**) were also analyzed by polarized FT-IR and UV–vis experiments after sample alignment by mechanical shearing. This process provided aligned arrays where the columnar phases were oriented with the columns parallel to the shearing direction[38]. Polarized UV–vis experiments of these samples revealed that in all cases the maximum absorption of the sample is reached with the polarizer parallel to the shearing direction (Supplementary Fig. 12). On the other hand, FT-IR spectra of the samples show the NH vibration at 3170 cm$^{-1}$, which indicates the presence of H-bonds (Supplementary Figs. 13–16)[38,43]. Importantly, the maximum intensity of the N–H vibration bands in compounds **2–4** was also observed with the polarizer parallel to the shearing direction by polarized FT-IR spectroscopy (Supplementary Figs. 14–16). These experiments unequivocally confirm that in the assemblies of PBIs **2–4**, the twisted rectangular PBI cores are oriented with their long

axis parallel to the columnar axis and establish intermolecular H-bonds (Fig. 1)[38].

To obtain detailed information on the assemblies of the LC PBIs **2–4**, WAXS patterns on aligned fibers were measured with the fiber axes standing approximately parallel and orthogonal with respect to the detectors tilt axis (Fig. 3).

Besides the discussed two-dimensional array of the columns, these experiments provide information about the direction of π-stacking of the mesogens and the periodical self-assembly along the columnar axes. Importantly, all the patterns show an additional diffuse intensity on the equator at around 4.1 Å. These patterns are characteristic for such PBI assemblies[38] and arise from the π–π distances between chromophores with orthogonal orientation with respect to the columnar axis (Fig. 3a, c, e). Furthermore, the WAXS patterns of the LC PBIs **2–4** present meridional and off-meridional reflections (Fig. 3b, d, f) that point to a periodic organization of PBIs along the columns.

PBI **2** (20 Col$_r$ 199 Col$_h$ 252 Iso) shows various layer lines along the meridian in the WAXS patterns at 160 °C in the Col$_r$ phase indicating the formation of a helical structure (Fig. 3a, b). The first reflection on the meridian corresponds to the axial translation subunits appearing at 13.8 Å[44], which is approximately the length of the perylene long axis. When this reflection is attributed to layer line $L = 14$, then all other meridional intensities are also perfectly positioned at layer lines $L = 28$, 44, and 54 (Fig. 3a, b and Supplementary Fig. 3). Additional diffuse reflections at small angles were demonstrated clearly in the previous work on a similar compound[38], supporting the helical packing arrangement. In the case of PBI **2**, these intensities are much weaker and are indicated with dashed green lines in Fig. 3a and Supplementary Fig. 4. Thus, the helical pitch and repeat contains 14 subunits in the Col$_r$ phase of PBI **2** and the assembly is classified as a $14_1$ helix with a pitch of 13.8 Å × 14 = 193.2 Å. Considering that the Col$_r$ phase of PBI **2** possesses four molecules per 13.8 Å columnar slice (Table 1), this assembly consists of a four-stranded helical structure (Fig. 1c). In contrast, three molecules fill the space of a columnar stratum of 13.8 Å in the Col$_h$ phase of PBI **2** at 200 °C (Table 1). The first reflection on the meridian of the WAXS pattern of the PBI **2** in its Col$_h$ phase appears at 13.8 Å. The best agreement with all observed layer lines

($L = 14$, 26, and 27) is obtained when this meridional reflection is assigned to layer line $L = 7$ (Supplementary Figs. 4 and 5). This matches with the formation of a triple-stranded $7_1$ helix with a helical pitch of 7 × 13.8 = 96.6 Å.

In a similar manner, *para*-functionalized PBI **3** exhibits three reflections on the meridian in its WAXS pattern (Fig. 3c, d) of the Col$_h$ phase that are attributed to layer lines $L = 7$, 9, and 28 (these layer lines can be clearly observed in Supplementary Fig. 7). Thus, PBI **3** forms a $7_1$ helix with a pitch of 14.1 Å × 7 = 98.7 Å. Considering that the number of molecules per strata for this Col$_h$ phase is three, PBI **3** forms triple-stranded helical columns with similar helix parameters to those observed for the Col$_h$ phases of PBI **2** and the previously described **MEH-PBI**[38]. In contrast, PBI **4** organizes into an intriguing Col$_h$ phase with only two molecules per columnar strata (Table 1). The WAXS pattern of PBI **4** at 224 °C shows various intensities along the meridian indicative of a helical arrangement. The meridional reflections appearing at 7.1, 6.9, and 3.6 Å could be best attributed to layer lines $L = 16$, 18, and 32, respectively (Fig. 3f and Supplementary Figs. 9 and 10). The axial translational subunit for this assembly is calculated to be 7.1 Å. This is only half of the length of a PBI molecule and can be rationalized with a centered structure along the double helix that causes the absence of the meridional 008 reflection assigned to the expected distance of the subunit of 14.2 Å. Accordingly, the assembly is an $8_1$ helix and the helical pitch and repeat is calculated to be 14.2 Å × 8 = 113.6 Å.

For a better understanding and illustration of these LC columnar structures, we generated geometry-optimized models of the helical structures of the PBIs **2**, **3**, and **4** with the program Accelrys Materials Studio 4.4. We assumed that the columns were formed of PBI strands in which the PBI cores are nanosegregated in the center of the column and surrounded by the flexible aliphatic chains in the outer part. To prepare the structural models we considered a molecular twist (~22° to 30°) of the PBI molecules[45], which is imposed by the four sterically demanding bay-substituents. The PBIs were then arranged according to the X-ray data to form H-bonded strands, which in turn form a double-stranded helix (helical pitch = 113.6 Å) for PBI **4**, a triple-stranded helix (helical pitch = 98.7 Å) for PBI **3** (Col$_h$ phase), or a quadruple-stranded helix (helical pitch = 193.2 Å) for PBI **2** (Col$_r$

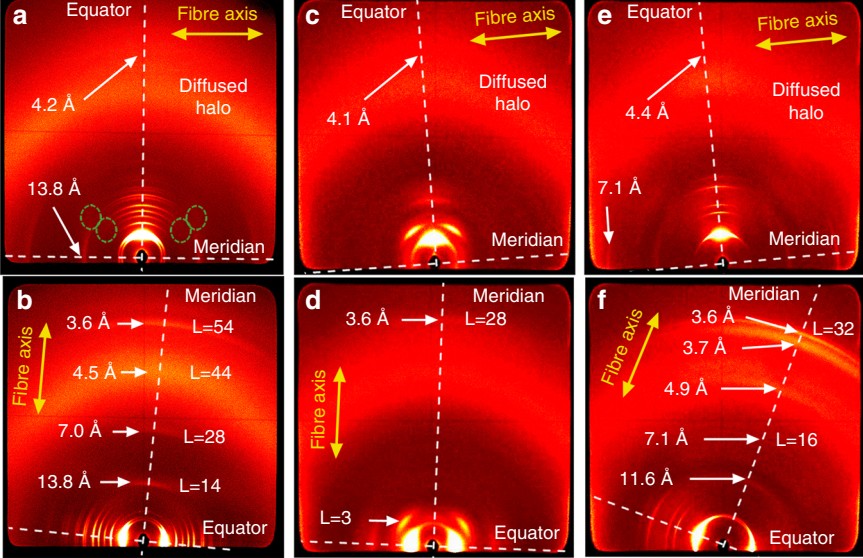

**Fig. 3** WAXS patterns of PBIs **2**, **3**, and **4**. WAXS diffraction pattern of aligned fibers of **a,b** PBI **2** at 160 °C, **c,d** PBI **3** at 180 °C, and **e,f** PBI **4** at 224 °C. The relative position of the fibers are indicated by yellow arrows and the meridian and equator by white dashed lines. The reflections on the equator are indexed according to Col$_h$ or Col$_r$ phases in Fig. 2. The reflections on the meridian that are attributed to layer lines are indicated as $L = X$. Dashed green lines indicate additional diffuse reflections at small angles

phase) (Supplementary Figs. 17–19). The corresponding helical assemblies were geometry-optimized by the force field Compass using the atom-based summation method in the corresponding hexagonal or centered rectangular unit cells. For all modeled structures (PBIs **2–4**), strongly negative non-bonding interaction energies have been achieved implying stable self-organized assemblies (Supplementary Figs. 17–19). The optimized structures of PBIs **2**, **3**, and **4** are shown in Fig. 4. Note that the centered structure along the column of PBI **4** is demonstrated by the shift of the fifth mesogen by $c/2$ and a 180° turn vs. the first mesogen within a single strand of the double-stranded assembly —the fifth mesogen of the red strand is then covered by the blue strand (Fig. 4, model c).

These modeled structures (Fig. 4) support that the PBIs stack hierarchically first in slipped and twisted π-aggregates that subsequently self-assemble into H-bonded helices. The forces driving these unprecedented assemblies are a combination of the nanosegregation of the PBI cores via H-bonded and slipped π–π interactions and the steric and space-filling demands of peripheral aromatic and aliphatic units[46]. It is noteworthy that the twist on the PBI cores, and thus the helical pitch, varies in each columnar assembly due to the different steric demands at the periphery of the mesogens. Remarkably, the substitution position of the dendrons at the phenoxy linker and the shape of the dendrons define: (a) the average twist of PBI cores, (b) the number of assembled strands per column, and (c) the helical pitch of the supramolecular structure. Thus, it is the steric crowding of the periphery, which restricts the number of aggregated perylene dyes and defines the precise helical morphology, while the slipped π-stacking of the axially twisted

perylene dyes and the H-bond interactions induces the helical assembly in the LC state.

In addition to the modeling, the fiber diffraction patterns of PBIs **2** (Col_r), **3** (Col_h), and **4** (Col_h) were simulated with CLEARER[47] utilizing the results of Materials Studio (Fig. 4). For the three different helices of PBIs **2**, **3**, and **4**, the simulation results are in good qualitative agreement with the experimental diffraction pattern as shown in Supplementary Fig. 20. Importantly, the reflections on the meridian attributed to the subunit axial translation and the equator fit with the experimental data. The weak additional reflections on the layer lines do not appear in experimental WAXS and MAXS, which is explained by the higher disorder in the LC material compared with the structure in a single unit cell used for simulation. It should be also noted that the pattern of PBI **4** indeed shows the absence of the meridional reflection at L = 8, and reveals instead a meridional signal located at the layer line 16. This simulation result is in very good agreement with the experiment (Supplementary Fig. 20) and confirms the molecular model.

**Spectroscopic investigations and exciton coupling.** The different organization of PBIs **1–4** offers a unique possibility for elucidating the influence of particular packing arrangements on dye aggregates' functional properties, in particular in terms of exciton coupling of the dyes' transition dipole moments in UV–vis absorption spectra. Accordingly, we compared the optical properties in the bulk state with the ones of their monomers in solution (Table 2, Fig. 5, Supplementary Fig. 11 and Supplementary Table 3). UV–vis absorption spectra of PBIs **1–4** as

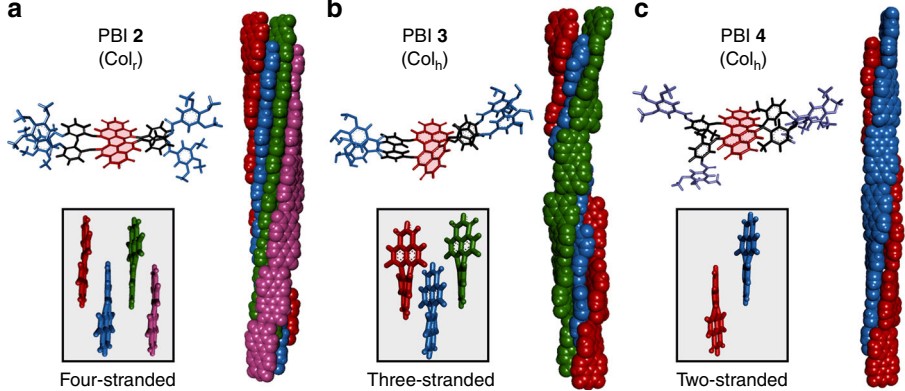

**Fig. 4** Optimized self-assembled structures of PBIs **2**, **3**, and **4**. Geometry-optimized monomers, tetramers/trimers/dimers, and four-/three-/two-stranded helices of PBI **2** (**a**), **3** (**b**), and **4** (**c**), respectively. The helicity in the columnar assemblies has been chosen to be (P) for graphical representation. Please note that PBIs **1–4** are racemic and therefore (P) and (M) helices may coexist in the LC phases

**Table 2 Optical properties of the PBIs in solution and solid state**

| PBI | Number of strands | Monomer (CH₂Cl₂)[a] | | | Solid state[b] | | |
|---|---|---|---|---|---|---|---|
| | | $\lambda_{max}$ (nm) | $\varepsilon$ (L mol⁻¹ cm⁻¹) | FWHM[c] (cm⁻¹) | $\lambda_{max}$ (nm) | FWHM[c] (cm⁻¹) | $\Delta\tilde{\nu}_{Agg-Mon}$ (cm⁻¹) |
| **1** | 1 | 562.5 | 41,400 | 1100 | 580.0 | 670 | −540 |
| **2** | 4 | 564.0 | 34,100 | 1350 | 625.5 | 1110 | −1740 |
| **3** | 3 | 569.0 | 41,600 | 1200 | 635.0 | 1120 | −1830 |
| **4** | 2 | 571.0 | 44,100 | 1200 | 647.5 | 850 | −2070 |

[a]CH₂Cl₂ ($c = 1.2$–$1.9 \times 10^{-5}$ M) at 25 °C
[b]Solution-sheared from methylcyclohexane solution ($c \sim 10^{-3}$ to $10^{-4}$ M) on quartz substrates
[c]FWHM was derived as twice the distance between the absorption maximum to the closest edge (here the red edge) at half maximum of the unsymmetrically shaped absorption bands to prevent falsification by the vibronic progression

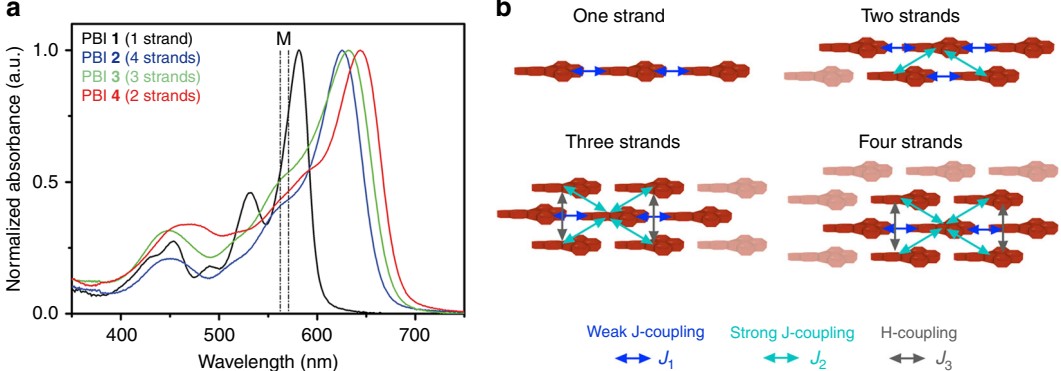

**Fig. 5** Solid state UV–vis spectra and exciton coupling schemes of the PBIs **1**–**4**. **a** Normalized UV–vis absorption spectra of PBIs **1** (black), **2** (blue), **3** (green), and **4** (red) in the solid state drop-casted from CHCl$_3$ solution are represented as solid lines (**1**: Cr; **2**, **3**, **4**: LC, $c_0 \sim 10^{-4}$ M) while the two vertical dashed lines indicate the position of the absorption maxima ($\lambda_{max}$) of their respective monomers in CH$_2$Cl$_2$ solution. **b** Schematic illustration of proposed J- and H-couplings between next neighbors in multi-stranded assemblies. Blue arrows indicate weak J-coupling ($J_1$), cyan arrows strong J-coupling ($J_2$), and gray arrows H-coupling ($J_3$) between PBIs, respectively

monomers were measured in CH$_2$Cl$_2$ ($c = 10^{-5}$ to $10^{-7}$ M), while for bulk materials, samples were prepared by drop-casting from chloroform ($c = 10^{-4}$ M) and solution shearing from methylcyclohexane (MCH) ($c = 10^{-4}$ M) on quartz substrates. The respective spectroscopic properties are summarized in Table 2. For the monomers almost identical spectral shapes with absorption maxima ($\lambda_{max}$) between 562 and 571 nm and molar extinction coefficients ($\varepsilon_{max}$) of about 40.000 L mol$^{-1}$ cm$^{-1}$, which are reasonable for tetraphenoxy bay-substituted PBIs[25], were observed.

For the solid state samples, a strong bathochromic shift of $\lambda_{max}$ to 625, 635, and 647 nm with respect to the monomers in solution was observed for PBIs **2**, **3**, and **4** (Table 2). The slight bathochromic shift to 580 nm with retained vibronic structure observed in PBI **1** supports that this compound does not pack in multiple strands due to the steric effects imparted by the *ortho*-substitution[40]. In contrast, the other PBIs show pronounced bathochromic shifts of $\lambda_{max}$ which is strongest for double-stranded PBI **4** (>2000 cm$^{-1}$) and decreases with increasing number of strands from PBI **3** (>1800 cm$^{-1}$; three strands) to the quadruple-stranded PBI **2** (>1700 cm$^{-1}$) (Fig. 5a).

In order to derive a structure–property relationship between the packing data from the X-ray experiments of the LC phases and the optical properties in the solid state, we apply Kasha's exciton theory based on the point-dipole approximation (PDA) excluding vibronic coupling. First, we deduced the geometrical parameters (center to center distance $r$, slip angle $\theta$, twist $\alpha$) from the self-assembled structures obtained from the Materials Studio models. Second, we calculated the exciton couplings, $J$, between nearest neighboring dyes within the PDA (Table 2, Supplementary Fig. 21, and Supplementary Tables 3 and 5)[48]. For all PBIs, the transition dipole moments ($\mu_{eg}$) were determined to be 6.6–7.1 D from the monomers' absorption spectra in CH$_2$Cl$_2$ (Table 2, Supplementary Fig. 11). These values are in good agreement with already published values for similar bay-substituted PBIs[43]. We found a weak J-coupling ($J_1$) of about −155 cm$^{-1}$ along the columnar axis (H-bonding direction) of almost equal size for all proposed structures (**1**–**4**) (Fig. 5b, Supplementary Table 5). Additionally, in all systems composed of multiple strands (**2**, **3**, and **4**) a significantly larger J-type coupling ($J_2$) of about −600 cm$^{-1}$ is present between slipped-π-stacked PBIs. This explains their much larger bathochromic shifts of $\lambda_{max}$ with respect to single-stranded PBI **1**. However, these two J-type couplings are partly compensated in the assemblies with three or four strands because of an additional H-type coupling ($J_3$) of

+370 and +435 cm$^{-1}$ for PBIs **2** and **3**, respectively, originating between chromophores in side-by-side arrangements of second next strands (Fig. 5b, Supplementary Table 5). Accordingly, with increasing number of self-assembled PBI strands more contributions of $J_3$ are present (Fig. 5b). In this regard, PBI **2** (4 strands) exhibits the smallest bathochromic shift of the series and PBI **4** (2 strands) the largest. Therefore, we envision that PBI **4** should be the most promising material for photonic applications.

For the purely H-bonded chain formed by PBI **1**, some further discussions are appropriate because here, in contrast to all other systems, a pronounced 0,1 vibrational transition at 519 nm prevails in addition to the dominant 0,0 transition at 562 nm (Fig. 5a). The existence of vibrational progression is understandable based on recent work of Spano that describes how competing excitonic and vibrational couplings influence the spectral shape of the aggregates' absorption bands[16]. Thus, in cases where the exciton coupling is dominant (as given in PBIs **2**, **3**, **4**), pronounced bathochromic shifts are observed with a loss of vibrational fine structure and application of the simple model of Kasha within PDA is sufficient[48]. However, in cases of weak exciton coupling compared to the vibrational coupling only very modest bathochromic shifts are observed and the main signature of the exciton coupling contribution is a redistribution of the intensity ratio of the 0,0 and the 0,1 vibrational bands (Supplementary Table 2)[26]. For PBI **1**, in full compliance with the expected J-type coupling for head-to-tail aligned H-bonded molecules, the intensity ratio for the absorption bands $A_{0,0}/A_{0,1}$ increases from the monomer to the thin films corroborating the presence of weak J-type coupling in H-bonded PBI chains (Supplementary Table 3). Further absorption and fluorescence spectroscopy studies are focused on aggregates of PBIs **1**–**4** in MCH in order to gain deeper insights into their intrinsic optical properties and supramolecular assemblies in solution.

## Discussion

In this work, we showed how the molecular engineering of PBI dyes can be used to design complex supramolecular assemblies with unprecedented packing patterns and intriguing optical properties. Tetraphenoxy-dendronized PBIs with two N–H functional groups are shown to form H-bonded linear chains with slipped-stack arrangements of the twisted PBI cores. This produces LC and crystalline columnar J-aggregates with the chromophores oriented parallel to the columnar axes. Remarkably, we could tune the composition of these columnar assemblies between

one, two, three, or four strands depending on the molecular design imparted by suitable dendron wedges. In this regard, the relative position of the dendrons at the phenoxy spacers of the PBI cores is the key factor that determines the respective packing structure by steric requirements (i.e., the twist of the perylenes, the helical pitch, and consequently the number of strands in a single column). Moreover, these structural features directly translate into functional properties due to the electronic coupling of the closely stacked dyes' transition dipole moments. Here we could show that the bathochromic shift of the absorption maximum in the aggregated state compared to the monomer is dependent on the number of self-assembled strands and affords the best J-aggregate (with purely negative couplings among neighbor molecules) for the double-stranded arrangement of PBI **4**. Strongly coupled J-aggregates are promising materials for photonic devices[49,50] and accordingly our future research is directed towards the investigation of these new materials in optical microcavities with the goal to elucidate the impact of structural organization and number of strands on photonic properties.

## Methods

**Synthesis of the PBIs**. The synthetic procedures for the preparation of PBIs **1**–**4** are described in Supplementary Methods.

**Wide-angle and middle-angle X-ray scattering (WAXS, MAXS)**. Temperature-dependent WAXS and MAXS investigations were performed on a Bruker Nanostar (Detector Vantec2000, Microfocus copper anode X-ray tube Incoatec). Aligned samples were prepared by fiber extrusion at 20–30 °C under the melting/clearing point of the respective compounds using a home-made mini-extruder. The fibers were transferred into Mark capillaries (Hilgenberg) and assembled in the heating stage of the Nanostar. WAXS experiments were performed at a sample-detector distance of 13 and 21 cm with the detector tilted by 14° upwards in order to study the angular range of $2\theta = 0.8°–28°$. In order to see the whole pattern, the experiments with the tilted detector have been performed with two different set-ups, i.e., orientations of the aligned fibers towards the tilt axis of the detector: (i) the fiber direction orthogonal with the tilt axis (standing fiber) showing meridional signals reminiscent of the helices and (ii) the fiber direction parallel with the tilt axis of the detector (lying fiber) visualizing the π–π distance. Additional MAXS studies have been carried out at a distance of 28 cm with a linear assembled detector, covering an angular range of $2\theta = 0.7°–12°$. Silver behenate was used as calibration standard for WAXS, MAXS. All X-ray data were processed and evaluated with the program datasqueeze (http://www.datasqueezesoftware.com/).

**Calculation of the number of molecules per columnar stratum**. The number of molecules $Z$ per columnar stratum was calculated using the following equation[51]: $Z = (\delta \cdot N_A \cdot V_{col\text{-}strat})/M$; where $\delta$ is the density, $M$ is the molecular mass, $N_A$ is the Avogadro's constant, and $V_{col\text{-}strat}$ is the volume of the columnar stratum. The density was extrapolated to be between 0.89-0.93 g cm$^{-3}$ (Supplementary Table 1) and the volume of the columnar stratum was calculated using to $V_{col\text{-}strat} = a^2 \cdot \sin 60° \cdot h$ for a Col$_h$ unit cell and $V_{col\text{-}strat} = a \cdot b \cdot h$ for a Col$_r$ unit cell. Because of the centered Col$_r$ phase (c2mm), the number of columns in the unit cell is two. We calculated the volume of one columnar stratum with a height $h$ of 13.8-14.2 Å, which is close to the length of the PBI long axis.

**UV–vis spectroscopy**. UV–vis absorption spectra in solution were recorded using a Perkin Elmer Lambda 35 spectrophotometer. The samples were prepared in spectroscopic grade solvent (Uvasol®, Merck, Hoehenbrunn, Germany) and quartz glass cuvettes were used. Extinction coefficients were calculated from Lambert–Beer's Law.

The solid-state studies were recorded with a Perkin Elmer Lambda 950 UV–Vis–NIR spectrophotometer equipped with an integrating sphere. The spectra were obtained in reflection mode. The samples were coated on Ulland quartz glass plates. For the polarized UV–vis measurements, a polarizer was inserted in between the light source and the sample. The aligned samples for polarized UV–vis measurements, were prepared on quartz glass substrates that were successively cleaned with chloroform, toluene, isopropanol, and acetone and heated to 70 °C. Then, 10 μL PBI solution ($10^{-4}$ to $10^{-3}$ M in MCH) was placed with an Eppendorf pipette between the substrate and the shearing plate (tilted angle = 15°) in a home-made shearing device and the solution was sheared at 0.083 mm s$^{-1}$ at 70 °C. The dichroic ratio of the aligned samples was calculated according to literature[52].

**FT-IR experiments**. Temperature-dependent and polarized FT-IR spectra were recorded with an AIM-8800 infrared microscope connected to a Shimadzu IRAffinity FT-IR spectrometer. The sample was prepared as a thin film on a KBr plate (thickness 2 mm), which was placed on a THMS600 heat stage with a Linkam TP94 controller. Polarization dependent FT-IR spectra were measured by using a precision automated polarizer (ZnSe) from PIKE Technologies. This includes the PIKE Technologies Motion Control Unit and AutoPro software. The aligned PBI samples for polarized FT-IR investigations were prepared by friction transfer of an extruded fiber on a hot KBr plate at 150 °C in the LC state.

**Modeling of the self-assembled structures**. The modeling and optimization of the LC phases were done by using Materials Studio 4.4 (for details see Supplementary Methods).

**Data availability**. All data supporting the findings in this study are available within the article and Supporting files, or are available from the corresponding authors upon reasonable request.

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

## Acknowledgments

We thank the Deutsche Forschungsgemeinschaft (DFG) for supporting this research (Grants LE 1571/7-1 and WU 317/18-1). P.L. thanks the Alexander-von-Humboldt foundation for a Georg-Forster postdoctoral stipend.

## Author contributions

S.H. and P.L. conducted the synthesis and characterization of the PBI molecules. S.H. characterized the materials in solid state and in solution (POM, DSC, WAXS/MAXS, UV–vis, FT-IR). M.L. and B.S. assisted and supervised the characterization of the LC materials. M.S. assisted and supervised the spectroscopic studies. F.W. conceived and supervised the whole work. All the authors contributed to the manuscript writing.

## Additional information

**Competing interests:** The authors declare no competing interests.

