## [Peer Review File · Nature Communications]

Reviewers' comments:

Reviewer #1 (Remarks to the Author):

Overall, this is a thorough paper from a high-quality group in the area. The work is well carried out, the paper is clear and the data are very interesting. However, to me, this seems too specialized for this journal. The work is detailed and thorough but I suspect will be of interest to a relatively small number of people who have sufficient interest to dig into the details covered here. The authors also hint at the end that this work opens up new avenues; it is not clear to me from this work how that is the case. The authors have found an interesting series, but it is not obvious from the current paper how one would take this information and design another series. On top of this, the results are linked to UV spectra, which while useful, have not been shown to link to any of the (for example) suggested photonic applications. I don't mean to be negative here, the paper is very interesting, but I think that this is more suited to a more specialized journal.

Regarding the H- and J- aggregates, the authors should probably consider referencing the following papers:

Wu et al. *Langmuir*, 2011, 27, 3074; Jancy and Asha, *J. Phys. Chem. B* 2006, 110, 20937; and especially when discussing the organization in the fibrous aggregates in bulk aggregates, Draper et al. *Chem* 2, 716–731

On page 8, the authors state that the thin films show shifted bands relative to the monomer in Fig. S10 – the caption to Fig. S10 states that this figure shows the monomer spectra only. It is also not clear to me how they authors can be sure that these show the spectra for the monomers - aggregation could still be highly prevalent even in dilute solution.

No concentration is specified in the caption to Fig. S11 or (as seems to be the case from the main text) that this is a thin film. Also, here, the statement seems to be that the monomer spectra were obtained in MCH. Fig. S10 provides monomer spectra in DCM. Are these identical in DCM and MCH? It seems that the data in DCM do not have absorption maxima in the positions suggested by the labels in Fig. S11 for MCH, which seems to imply a degree of aggregation perhaps in at least one of these solvents. Further, the authors compare the spectra in dilute solution and thin films - what happens at higher concentrations in solution?

Is a density of 1g/cm³ reasonable? On what basis?

I can't really judge the modelling and geometry optimization work effectively. However, no optimization is provided for PBI 1. Why not? This would presumably provide more confidence considering that the data should match the packing of 1 as determined by pXRD.

The conclusions end by suggesting that this work opens up new avenues for the design of strongly coupled J-aggregates. It's not clear to me from reading this work how one would use the current work to design further examples that exhibit these properties. It would be good if the authors spelt this out more clearly. Can any design elements be described?

Reviewer #2 (Remarks to the Author):

Manuscript Number: NCOMMS-18-06829

Title: Self-assembly of Multi-Stranded Perylene Dye J-Aggregates in Columnar Liquid Crystalline Phases

Recommendation: Publish after minor revision.

Comments:

The manuscript reports a series of crystalline and liquid-crystalline perylene bisimides that self-assemble into multi-stranded (two, three and four strands) aggregates with predominant J-type exciton coupling. The supramolecular structure and optical property of the complex were investigated via different experimental techniques such as differential scanning calorimetry, wide-angle and middle-angle X-ray scattering (WAXS, MAXS), molecular simulation and UV-Vis spectroscopy. I suggest accepting the paper after minor revision. My minor reservations to the manuscript are listed below.

1. There are several typos in this manuscript. Please check them.

For example:

P5 L101: PBIs 2-4 where (were) obtained as waxy dark blue solids.

P6 L114: More details about these calculations (are shown) in the Supplementary Information.

P12 L256 Accelrys Material (Materials) Studio.

2. Page 11, the authors stated "The first reflection on the meridian corresponds to the axial translation subunits appearing at 13.8 Å, which is approximately the length of the perylene long axis. When this reflection is attributed to layer line L = 14, then all other meridional intensities are also perfectly positioned at layer lines L = 28, L = 44 and L = 54 (Figs 3a,b and Supplementary Fig. 3)." However, it seems there is no distinct low-angle diffraction arcs on the meridian and in the quadrants which similar to the 2D XRD pattern in reference 39, so the four culminant diffraction arcs in Fig 3(a, b) can also be indexed as (001), (002), (003) and (004), respectively. Similarly, the meridional diffraction arcs (13.8 Å, 4.7 Å and 3.6 Å) in Supplementary Figure 4 can also be indexed as (001), (003) and (004). In this case, the Colr phase and Colh phase of PBI 2 may not a helical structure. The problem probably lies in the low-resolution 2D XRD image. It is better to provide an appropriate interpretation or clearer 2D XRD pattern.

3. Page 8, the authors stated "The lower temperature Colr phase of PBI 2 has four molecules per columnar slice while the higher temperature Colh phase consists of three molecules per slice (Table 1)." This thermally-induced phase transition from four-stranded helical structure to three-stranded helical structure is interesting and unanticipated. Is it possible to get a qualitative insight into this surprising physical result?

4. In Supplementary Figure 2, Supplementary Figure 4 (left), Supplementary Figure 6, Supplementary Figure 8 and Supplementary Figure 19 (right), the fiber direction (indicated by yellow arrows) and equator/meridian direction are not consistent with that in Fig. 3, please check whether the fiber direction and equator/meridian direction are wrong. If the directions are correct, it is better to provide an appropriate interpretation why the stranded helical columns are not parallel to the fiber direction.

5. When the density was assumed to be 1 g cm⁻³, the number of molecules per columnar stratum seems to be about 3.4, 3.5 and 2.2 for Colh phase of PBI2, PBI3 and PBI4, respectively. Please check the number of molecules. It should be more precise to calculate the molecules number based on the experimental density.

Reviewer #3 (Remarks to the Author):

The authors demonstrate an interesting method for controlling intermolecular packing via small variations of molecular design; in particular, the position (ortho, meta, para) at which a large dendron is attached to a bay-functionalized phenoxy group on a perylene diimide (PDI)

chromophore. After a thorough characterization based on wide-angle x-ray scattering, IR and UV-Vis spectroscopies, etc the authors show that the molecular variants packed in very different structures; 1) a triclinic crystal with largely isolated single, head-to-tail PDI strands and 2-4 three liquid crystals forms hosting 2, 3 or 4 PDI strands. The UV-Vis spectra, and in particular the position of the main absorption peaks were in qualitative agreement with Kasha's theory for J and H-aggregates based on point-dipole calculated Coulomb couplings. The single-stranded structure also abided by the Spano theory for vibronic intensity redistribution in J-aggregates. Overall, the paper shows novel results that are of substantial interest to the organic electronics community. There are precious few methods available to achieve control over intermolecular packing and the present one shows significant potential. Just a few suggestions/criticisms

1) Check grammar! The very first sentence in the abstract cannot be understood at all!

2) As the main message of the paper critically depends on knowing precisely the number of strands in each LC column, more discussion should be devoted to the method's accuracy - for example, on pg.8. Also, are there other studies which use the columnar slice method to determine the number of molecules per slice? Is this a well-established method?

2) More discussion should be devoted to the impact of the PDI twist on the direction of the molecular transition dipole moment (tdm). Does the twist impact the relative alignment between the tdms on neighboring molecules? Moreover, in the amphiphilic cyanine nanotube studies of Knoester and coworkers the chiral nature of the packing resulted in a tdm component along the tube axis as well as tangent the surface leading to strongly anisotropic polarized absorption spectra. Is the situation here similar? It appears the the LCs are also helical with a dominant component along the axis but should there also be an orthogonal component? Perhaps its too small to resolve?

3) Is there anything to be said as to why a particular conformer (PBI 1-4) resulted in two, three or four stranded LCs? Is there a design paradigm here? For example, what would be necessary to design a 5-stranded LC?

Point-to-Point Response to Reviewers' comments:

General: We thank all reviewers for their valuable comments that helped us to improve our manuscript.

Reviewer #1 (Remarks to the Author):

Comment 1: Overall, this is a thorough paper from a high-quality group in the area. The work is well carried out, the paper is clear and the data are very interesting. However, to me, this seems too specialized for this journal. The work is detailed and thorough but I suspect will be of interest to a relatively small number of people who have sufficient interest to dig into the details covered here. The authors also hint at the end that this work opens up new avenues; it is not clear to me from this work how that is the case. The authors have found an interesting series, but it is not obvious from the current paper how one would take this information and design another series. On top of this, the results are linked to UV spectra, which while useful, have not been shown to link to any of the (for example) suggested photonic applications. I don't mean to be negative here, the paper is very interesting, but I think that this is more suited to a more specialized journal.

Answer 1: Our work is of interest for chemists and for materials scientists. It is very typical for these fields that first a new molecule or new material is introduced and only later applications come up. For instance when Chandrasekhar introduced the first example for a columnar liquid crystal in 1977 no-one could have imagined thousands of examples to be synthesized in the following thirty years with significant impact on supramolecular chemistry and organic electronics (*Chem. Rev.* **2016**, *116*, 1139-1241). However, whilst useful for charge carrier transport, the typical arrangement of the p-systems in columnar liquid crystals with H-type coupling of the dyes' transition dipole moments and concomitant loss of fluorescence is not suitable for photonic applications. Only in 2017 we introduced a first example for a new type of columnar liquid crystals (*Angew. Chem. Int. Ed.* **2017**, *56*, 2162) where the dyes are organized differently, leading to J-type coupling and fluorescence. Immediately afterwards, successful implementation of this material in microcavities demonstrated its usefulness for photonics applications (*Appl. Phys. Lett.* **2017**, *110*, 201113-1–201113-4; *Adv. Optical Mater.* **2017**, 1700523; *ACS Photonics* **2018**, *5*, 90–94). Accordingly we do not expect that the here introduced new materials will remain laboratory curiosities. Because here it is shown how dye arrangement can be controlled and excitonic properties can be adjusted. This will be crucial for the design of desirable soft materials for photonic devices.

Comment 2: Regarding the H- and J- aggregates, the authors should probably consider referencing the following papers: Wu et al. *Langmuir*, 2011, *27*, 3074; Jancy and Asha, *J. Phys. Chem. B* 2006, *110*, 20937; and especially when discussing the organization in the fibrous aggregates in bulk aggregates, Draper et al. *Chem* 2, 716–731

Answer 2: The most relevant references to competing H- and J-type aggregation (ref. 17-21) are the reviews cited in ref. 17, 18, 20 and 24. The other two references (19, 21) are first examples for the class of perylene bisimide dyes. If we include references to the work of Wu (*Langmuir* **2011**) as well as Jancy and Asha (*JPCB* **2006**) we should include at least another 20 equally relevant references on other related work on perylene bisimides which does not comply with the guidelines of *Nature*

Communications (max. 70 references). Thus, we decided to only add the most recent reference as a new reference 22 (and to remove the previous ref. 25 to our own work because ref. 26 is here sufficient):

22. E. R. Draper, B. J. Greeves, M. Barrow, R. Schweins, M. A. Zwijnenburg, D. J. Adams, *Chem* **2017**, *2*, 716-731.

Comment 3: On page 8, the authors state that the thin films show shifted bands relative to the monomer in Fig. S10 – the caption to Fig. S10 states that this figure shows the monomer spectra only. It is also not clear to me how they authors can be sure that these show the spectra for the monomers - aggregation could still be highly prevalent even in dilute solution.

Answer 3: Our group has worked for more than two decades on perylene bisimides bearing four phenoxy substituents in bay area (see e.g. F. Würthner, *Pure Appl. Chem.* **2006**, *78*, 2341–2349: Bay-substituted perylene bisimides: Twisted fluorophores for supramolecular chemistry). Accordingly, we know how monomer spectra for these dyes look like including the position of the absorption maxima and the patterns of vibronic coupling. Furthermore, we have spent a lot of work in understanding the solvent dependent aggregation of perylene bisimide dyes. The most in-depth study (Z. Chen, B. Fimmel, F. Würthner, *Org. Biomol. Chem.* **2012**, *10*, 5845–5855: Solvent and substituent effects on aggregation constants of perylene bisimide pi-stacks – a linear free energy relationship analysis) shows that aggregation is almost impossible to achieve in dichloromethane. Even for the by far more strongly aggregating core-unsubstituted perylene bisimides, binding constants in dichloromethane are not exceeding 1000 M^{-1} . As a consequence, no aggregation will be seen at concentrations of 10^{-5} M as used for UV/Vis absorption experiments in Figure S10.

Comment 4: No concentration is specified in the caption to Fig. S11 or (as seems to be the case from the main text) that this is a thin film. Also, here, the statement seems to be that the monomer spectra were obtained in MCH. Fig. S10 provides monomer spectra in DCM. Are these identical in DCM and MCH? It seems that the data in DCM do not have absorption maxima in the positions suggested by the labels in Fig. S11 for MCH, which seems to imply a degree of aggregation perhaps in at least one of these solvents. Further, the authors compare the spectra in dilute solution and thin films - what happens at higher concentrations in solution?

Answer 4: As indicated in the main text (“...solid state UV/Vis spectra (Supplementary Figs. 11)...”) the polarized UV-Vis spectra were obtained for thin films as also stated in the methods parts. We have now also clarified this in the caption by adding “thin films of”.

Because indeed in our manuscript no monomer spectra are shown for the solvent MCH (these spectra can be obtained at elevated temperature and at low concentration) we have now modified Fig. S11 with the dashed lines for monomers in dichloromethane (as given in Fig. S10). Typically, the spectral shift observed for monomeric perylene bisimide dyes caused by the solvent is rather small (e.g. absorption maxima for monomeric PBI **3** are located at 570 nm in dichloromethane and at 560 nm in methylcyclohexane).

Comment 5: Is a density of 1 g/cm^3 reasonable? On what basis?

Answer 5: In general, the assumption of 1 g/cm^3 is reasonable for these kind of molecules since mesogens consist usually of about 50-60 m% aliphatic chains with a density below 1 g/cm^3 and 40-50 m% aromatics with a density higher than 1 g/cm^3 . Temperature-dependent densities of liquid crystals have been measured for some molecules by dilatometry, see for example (*Chem. Eur. J.* **2001**, *7*, 1006–1013; *J. Am. Chem. Soc.* **2004**, *126*, 15258 – 15268; *Chem. Eur. J.* **2008**, *14*, 3562 – 3576.). However, in our work, as also pointed out by reviewers 2 and 3, the accuracy of the calculation of the number of strands per columnar stratum is crucial. Therefore, following the advice of all three reviewers, we have performed density measurements on compounds **2** and **4** by the buoyancy method and we could determine minimum densities at $20 \text{ }^\circ\text{C}$ for PBI **2**: $\delta = 1.018 \pm 0.006 \text{ g/cm}^3$ and PBI **4**: $\delta = 1.028 \pm 0.009 \text{ g/cm}^3$. These densities give the molecular volume ($V_{\text{mol}} = M/\delta/N_A$), that is the sum of volume of the aromatic units and the aliphatics. The temperature-dependence of the aliphatic volume is known (*J. Am. Chem. Soc.* **2004**, *126*, 15258 – 15268). The temperature-dependence of the aromatic scaffold, we calculated with the help of recently published report on a model perylene (*CrystEngComm* **2016**, *18*, 4787–4798). This allowed us to extrapolate the molecular volumes and consequently the densities for all temperatures, at which we performed the X-ray experiments. Subsequently, the number of strands was calculated. This is described in detail in the Supplementary Information (see also reviewer 2; answer to comment 12).

Comment 6: I can't really judge the modelling and geometry optimization work effectively. However, no optimization is provided for PBI 1. Why not? This would presumably provide more confidence considering that the data should match the packing of 1 as determined by pXRD.

Answer 6: In contrast to the other three PBIs, PBI **1** does not form a liquid crystal but a crystalline phase. Thus, PBI **1** is not viscous and cannot be properly aligned to get sufficient information for modelling (see our comment in Table 1 "The number of molecules per repeating unit of the strand in the crystalline phase could not be calculated from the X-ray data, but FT-IR and UV-Vis spectroscopy suggest that the assembly consists of single strands (vide infra).") This is also pointed out in the main text of the manuscript: "In the case of PBI **1**, the exact arrangement of the dyes could not be determined by the X-ray studies, but UV-Vis and FT-IR spectroscopy experiments are only consistent with a hydrogen-bond directed assembly based on a single strand."

Comment 7: The conclusions end by suggesting that this work opens up new avenues for the design of strongly coupled J-aggregates. It's not clear to me from reading this work how one would use the current work to design further examples that exhibit these properties. It would be good if the authors spelt this out more clearly. Can any design elements be described?

Answer 7: Following the reviewer's advice, we changed the last sentence:

"Strongly coupled J-aggregates are promising materials for photonic devices^{50,51} and accordingly our future research is directed towards the investigation of these new materials in optical microcavities with the goal to elucidate the impact of structural organization and number of strands on photonic properties."

In addition we like to point out that already the solid state UV/Vis spectra shown in Fig. 5 and our analysis by exciton coupling provide important clues, i.e. that the double stranded arrangement of PBI 4 is most promising for applications. This suggests that PBI 4 should afford a better material compared to the previously investigated derivative of PBI 2. However, because many additional parameters influence the materials' behavior in the bulk and the processing conditions for each material need to be optimized we do not like to speculate too much at the end of the given manuscript that is exclusively devoted to the supramolecular materials design.

Reviewer #2 (Remarks to the Author):

Comment 8: The manuscript reports a series of crystalline and liquid-crystalline perylene bisimides that self-assemble into multi-stranded (two, three and four strands) aggregates with predominant J-type exciton coupling. The supramolecular structure and optical property of the complex were investigated via different experimental techniques such as differential scanning calorimetry, wide-angle and middle-angle X-ray scattering (WAXS, MAXS), molecular simulation and UV-Vis spectroscopy. I suggest accepting the paper after minor revision. My minor reservations to the manuscript are listed below.

There are several typos in this manuscript. Please check them.

For example:

P5 L101: PBIs **2-4** where (were) obtained as waxy dark blue solids.

P6 L114: More details about these calculations (are shown) in the Supplementary Information.

P12 L256 Accelrys Material (Materials) Studio.

Answer 8: We have corrected the typos.

Comment 9. Page 11, the authors stated "The first reflection on the meridian corresponds to the axial translation subunits appearing at 13.8 Å, which is approximately the length of the perylene long axis. When this reflection is attributed to layer line L = 14, then all other meridional intensities are also perfectly positioned at layer lines L = 28, L = 44 and L = 54 (Figs 3a,b and Supplementary Fig. 3)." However, it seems there is no distinct low-angle diffraction arcs on the meridian and in the quadrants which similar to the 2D XRD pattern in reference 39, so the four culminant diffraction arcs in Fig 3(a, b) can also be indexed as (001), (002), (003) and (004), respectively. Similarly, the meridional diffraction arcs (13.8 Å, 4.7 Å and 3.6 Å) in Supplementary Figure 4 can also be indexed as (001), (003) and (004). In this case, the Colr phase and Colh phase of PBI 2 may not a helical structure. The problem probably lies in the low-resolution 2D XRD image. It is better to provide an appropriate interpretation or clearer 2D XRD pattern.

Answer 9: To clarify this issue we measured again some X-ray diffraction pattern of PBI 2. Here we got aware that we used in our submission uncorrected temperatures (set temperature of the XRS heating stage) for the XRS measurements and for Table 1 peak temperatures instead of standard onsets values! Therefore we completely checked and revised the given temperature values of XRS and DSC by using corrected temperatures and onsets (in the SI the DSC traces still contain the peak values and in addition now also the onsets. Note that this does not affect any XRS evaluation data; this data is all evaluated correctly).

In reference 39 we interpreted the diffraction pattern of the triple helical liquid crystal as superposition of the fiber diffraction pattern (generated by Clearer from the *Materials Studio* model) and the pattern of a single triple helix (form factor generated by the program Helix). This can be rationalized by the fluid nature and disorder of the liquid crystal. Here all small angle intensities are rather diffuse but clearly visible. In our present work the mesogens are similar and the XRS pattern show also most of the features than MEH-PBI in reference 39, but even the strongest meridional signals are not as intense than for MEH-PBI.

In order to show better the small angle diffuse intensities we repeated the measurements of PBI **2** at 160 and 200 °C with an increase of the time for recording the diffraction pattern. Especially the measurement at 200 °C in the Col_h phase (new supplementary Figure 4) shows now clearly the diffuse intensities, which are highlighted by green broken lines. Also the integrations along the equator and the meridian have been exchanged by the new measurements (supplementary Figure 5). Again the integration along the meridian shows now a diffuse intensity corresponding to layer line 6. For the new measurements, we payed especially attention to the correct calibration for integration along the meridian and fitted the visible meridional reflections to layer lines L = 6,7,14, 26 and 27.

New measurements at 200 °C of Supplementary Figure 4: PBI **2** cross shaped diffuse intensity at small angles confirming the helical arrangement. (Only half of the cross is visible owing to the tilted detector set-up).

In addition to the new Figures the following sentence has been added in our manuscript:

“Additional diffuse reflections at small angles were demonstrated clearly in the previous work on a similar compound,³⁹ supporting the helical packing arrangement. In the case of PBI **2**, these intensities are much weaker and are indicated with dashed green lines in Fig. 3a and Supplementary Fig. 4.”

The WAXS pattern of the Col_r phase at 160 °C (new Figures 3a,b and Supplementary Figure 3) exhibit also the weak diffuse intensity. The fit of the meridional reflections remains at layer-lines 14, 28, 44 and 54 remains (Supplementary Figure 3). The interpretation with layer lines L = 1 – 4 is not valid since the meridional reflections would not fit to these lines (see figure below):

Eventually, we included a Supplementary Figure 8 (right) showing an X-ray pattern at 163 °C for PBI **4**, in which the diffuse signals and layer lines are clearly visible. We highlighted the absence of the meridional reflection at layer-line 8 and the presence of the signal at layer line 16, supporting the centered arrangement.

Apart from these experimental evidences, there are many arguments in favor of a helix. The molecules possess bulky side groups. If they would stack in a way that only layer lines 1-4 were visible, then this would be a structure reminiscent of layers along the column. For steric and space-filling reasons two neighboring aggregates would need to turn by 90 ° and the next mesogenic aggregate turn back to 0° (any other angle would not disperse the peripheral building blocks uniformly around the column or if turned in the same direction and not back and forth it would generate a helix). Even the 90 ° angle can be considered as a special case of a helix with four molecules in the helical pitch and repeat unit. However, a 90° turn of the mesogens along the column back and forth generates a structure in which the real repeat unit would be 28 Å and not 14 Å. This is a further reason that does not support the interpretation with $L=1-4$. Eventually, the best packing in a subunit built from 2, 3 or 4 perylene bisimides is the slipped arrangement of the perylene chromophores. The J-couplings of these aggregates are evident from the UV/Vis spectra. These aggregates are already small helical units and can be stacked by nanosegregation and space filling only by the continuation of the helix presented in our models. This is also the only way that enables the aggregates to optimize the hydrogen bonding between the perylene units.

Comment 10: Page 8, the authors stated “The lower temperature Colr phase of PBI 2 has four molecules per columnar slice while the higher temperature Colh phase consists of three molecules per slice (Table 1).” This thermally-induced phase transition from four-stranded helical structure to three-stranded helical structure is interesting and unanticipated. Is it possible to get a qualitative insight into this surprising physical result?

Answer 10: Indeed, this is an interesting result. However, this is based simply on the fact that the chain volume increases more strongly with temperature than the volume of the aromatics. Thus, the steric crowding at the periphery increases. As a consequence, the steric crowding is compensated by the reduction of the number of mesogens in the columnar slice. This will be an issue of more detailed investigations in the future.

Comment 11. In Supplementary Figure 2, Supplementary Figure 4 (left), Supplementary Figure 6, Supplementary Figure 8 and Supplementary Figure 19 (right), the fiber direction (indicated by yellow arrows) and equator/meridian direction are not consistent with that in Fig. 3, please check whether the fiber direction and equator/meridian direction are wrong. If the directions are correct, it is better to provide an appropriate interpretation why the stranded helical columns are not parallel to the fiber direction.

Answer 11: We checked all the diffraction patterns again (see also our answer 9) and improved the consistency of the lines and arrows indicating meridian, equator and the fiber directions. Additionally, we added to the methods section the following sentence:

“In order to see the whole pattern the experiments with the tilted detector have been performed with two different set-ups, i.e. orientations of the aligned fibres towards the tilt axis of the detector: (i) the fibre direction orthogonal with the tilt axis (standing fibre) showing meridional signals reminiscent of the helices. (ii) the fibre direction parallel with the tilt axis of the detector (lying fibre) visualizing the π - π distance.

Comment 12. When the density was assumed to be 1 g cm^{-3} , the number of molecules per columnar stratum seems to be about 3.4, 3.5 and 2.2 for Colh phase of PBI2, PBI3 and PBI4, respectively. Please check the number of molecules. It should be more precise to calculate the molecules number based on the experimental density.

Answer 12: As already pointed out above (see Comment/Answer 5) temperature-dependent densities of such materials are not easy to obtain. As described in our Answer 5 above, and following work of Bertrand Donnio (Strasbourg) published in *CrystEngComm*, 2016, 18, 4787–4798, we could now obtain densities for all temperatures at which we performed the X-ray experiments. Details are given in the Supplementary Table 1. Now we calculated the number of molecules per columnar stratum for the experimental height of the unit cell (13.8-14.2 Å) and not the previously used constant average height of 14 Å. The results are:

PBI 2 (Colr, 160°C) 0.930 g/cm³, 3.7 ± 0.3 molecules /per stratum; [old value at a density of 1 g/cm³: 4.1 molecules/stratum]

PBI 2 (Colh, 200 °C) 0.993 g/cm³, 3.0 ± 0.2 molecules / stratum; [old value at a density of 1 g/cm³: 3.4 molecules/stratum]

PBI 3 (Colh, 180 °C) 0.916 g/cm³, 3.2 ± 0.2 molecules / stratum; [old value at a density of 1 g/cm³: 3.4 molecules/stratum]

PBI 4 (224 °C) 0.895 g/cm³, 2.0 ± 0.1 molecules / stratum; [old value at a density of 1 g/cm³: 2.2 molecules/stratum]

Assuming rather large errors of ± 1 Å for the cell parameters a_{hex} and a_{rec} , b_{rec} , ± 0.5 Å for the height of the stratum and 0.02 g/cm^3 for the density, the error propagation results in the maximum deviation for the calculation of the numbers of molecules per columnar stratum of 0.1-0.3 molecules. The new calculations are nevertheless in good agreement with our previous claim and support now strongly the double, triple and quadruple stranded models.

Reviewer #3 (Remarks to the Author):

Comment 13: The authors demonstrate an interesting method for controlling intermolecular packing via small variations of molecular design; in particular, the position (ortho, meta, para) at which a large dendron is attached to a bay-functionalized phenoxy group on a perylene diimide (PDI) chromophore. After a thorough characterization based on wide-angle x-ray scattering, IR and UV-Vis spectroscopies, etc the authors show that the molecular variants packed in very different structures; 1) a triclinic crystal with largely isolated single, head-to-tail PDI strands and 2-4) three liquid crystals forms hosting 2, 3 or 4 PDI strands. The UV-Vis spectra, and in particular the position of the main absorption peaks were in qualitative agreement with Kasha's theory for J and H-aggregates based on point-dipole calculated Coulomb couplings. The single-stranded structure also abided by the Spano theory for vibronic intensity redistribution in J-aggregates. Overall, the paper shows novel results that are of substantial interest to the organic electronics community. There are precious few methods available to achieve control over intermolecular packing and the present one shows significant potential. Just a few suggestions/criticisms.

Check grammar! The very first sentence in the abstract cannot be understood at all!

Answer 13: We did our best (see also Answer 8) and changed the first sentence. Also a postdoc in our group who is a native speaker helped us to improve the language in the whole manuscript.

Comment 14: As the main message of the paper critically depends on knowing precisely the number of strands in each LC column, more discussion should be devoted to the method's accuracy - for example, on pg.8. Also, are there other studies which use the columnar slice method to determine the number of molecules per slice? Is this a well-established method?

Answer 14: We have answered this comprehensively already for reviewer 2 (answer to Comment 12). This is a well established method to estimate the numbers of molecules per columnar stratum. The maximum error can be estimated by error propagation to be not more than 0.1-0.3 molecules.

Comment 15: More discussion should be devoted to the impact of the PDI twist on the direction of the molecular transition dipole moment (tdm). Does the twist impact the relative alignment between the tdm's on neighboring molecules? Moreover, in the amphiphilic cyanine nanotube studies of Knoester and coworkers the chiral nature of the packing resulted in a tdm component along the tube axis as well as tangent the surface leading to strongly anisotropic polarized absorption spectra. Is the situation here similar? It appears the LCs are also helical with a dominant component along the axis but should there also be an orthogonal component? Perhaps its too small to resolve?

Answer 15: The situation here is different from the one in the mentioned helical cyanine dye tubes investigated by Knoester (and also by researchers from Berlin). In our case the perylene bisimide dyes' transition dipole moment is oriented along the axis of the aggregate chain which is also the axis of the helical twist (see Fig. 4). According to the exciton chirality method of Harada and Nakanishi, there is no (or only weak) exciton coupling to be expected for this arrangement of the transition dipole moments. This issue has been discussed in a previous work from our group on the self-assembly of structurally related perylene bisimide dyes in solution (ref. 26).

Comment 16: Is there anything to be said as to why a particular conformer (PBI 1-4) resulted in two, three or four stranded LCs? Is there a design paradigm here? For example, what would be necessary to design a 5-stranded LC?

Answer 16: The packing arrangement can be understood based on two important concepts for the packing in condensed phases: Nanosegregation and space filling. With the substitution pattern in the bay positions of the perylene dyes we are able to influence the steric crowding at the periphery and thus only a maximum number of 2,3 or 4 perylenes can stack together in a repeating unit (aggregate) without an overcrowding of the aliphatic chains. The steric crowding certainly can be controlled in the future when using mixtures of molecules bearing only 1 or 2 aliphatic chains per aromatic unit in the periphery with the present derivatives. With this strategy (reduction of the steric crowding) it should be possible to increase the number of molecules in the columnar stratum to numbers exceeding 4. In order to highlight this issue, we added the sentence:

“Thus, it is the steric crowding of the periphery, which restricts the number of aggregated perylene dyes and defines the precise helical morphology, while the slipped π -stacking of the axially twisted perylenes and the H-bond interactions induces the helical assembly in the liquid-crystalline state.”

REVIEWERS' COMMENTS:

Reviewer #1 (Remarks to the Author):

The authors have revised the manuscript in light of the comments of the referees and added more data, explanation and references. Overall, I'm still not convinced that this will have the impact that the authors seem to think, but the paper is interesting and I am happy that the points have been addressed and that the paper is suitable for publication.

Reviewer #2 (Remarks to the Author):

I have re-reviewed the manuscript and find all changes acceptable. Thus I feel it is now suitable for publication.

Reviewer #3 (Remarks to the Author):

I believe the authors have adequately addressed all of the reviewer concerns and is now ready for publication.

Point-by-point response to referee comments

Manuscript: NCOMMS-18-06829A

REVIEWERS' COMMENTS:

Reviewer #1 (Remarks to the Author):

The authors have revised the manuscript in light of the comments of the referees and added more data, explanation and references. Overall, I'm still not convinced that this will have the impact that the authors seem to think, but the paper is interesting and I am happy that the points have been addressed and that the paper is suitable for publication.

Reviewer #2 (Remarks to the Author):

I have re-reviewed the manuscript and find all changes acceptable. Thus I feel it is now suitable for publication.

Reviewer #3 (Remarks to the Author):

I believe the authors have adequately addressed all of the reviewer concerns and is now ready for publication.

Answer: We are delighted by the statements of the referees recommending publication of our work and are very grateful to the reviewers for their comments during the reviewing process, which helped us to improve our manuscript.